# Modification of Poly(4-methyl-2-pentyne) in the Supercritical Fluid Medium for Selective Membrane Separation of CO_2_ from Various Gas Mixtures

**DOI:** 10.3390/polym12112468

**Published:** 2020-10-24

**Authors:** Viktoriya Polevaya, Anton Vorobei, Andrey Gavrikov, Samira Matson, Olga Parenago, Sergey Shishatskiy, Valeriy Khotimskiy

**Affiliations:** 1A.V. Topchiev Institute of Petrochemical Synthesis RAS, Leninsky pr., 29, 119991 Moscow, Russian; matson@ips.ac.ru (S.M.); hotimsky@ips.ac.ru (V.K.); 2Kurnakov Institute of General and Inorganic Chemistry RAS, Leninsky pr., 31, 119991 Moscow, Russian; vorobei@supercritical.ru (A.V.); penguin1990@yandex.ru (A.G.); oparenago@scf-tp.ru (O.P.); 3Institute of Polymer Research, Helmholtz-Zentrum Geesthacht, Max-Planck-Strasse 1, 21502 Geesthacht, Germany

**Keywords:** polyacetylenes, surface modification, supercritical fluids, highly permeable membranes, gas separation, quaternization, CO_2_ separation

## Abstract

The modification of highly permeable films of brominated 1,2-disubstituted polyacetylene, poly(4-methyl-2-penthyne), via incorporation of in situ formed butylimidazolium bromide is reported for the first time. Principal possibility and efficiency of supercritical CO_2_ and CHF_3_ use as reaction media for the corresponding process, namely for quaternization of butylimidazole by brominated polymer are revealed. As a result, we prepared new membrane materials possessing promising properties such as stability toward organic solvents, good mechanical properties and significantly improved CO_2_-selectivity while maintaining gas permeability at high values. Comparative analysis of the results allowed us to determine content and conditions for the incorporation of butylimidazolium groups optimal for most efficient separation of CO_2_ from industrial gas mixtures. These results are of interest for the designing of new CO_2_ selective membranes.

## 1. Introduction

The problem of CO_2_ separation from industrial gas streams of various compositions is among most prominent ecological challenges due to well-known greenhouse effect of carbon dioxide provoking considerable climate changes. Currently, various methods are intensively investigated in this respect, e.g., chemical and physical sorption, and cryogenic and membrane separation. The most widespread industrial method is absorption by organic solvents, e.g., amines. However, further development of this approach is restricted by high energy consumption during solvent regeneration, poisoning of amines with minor flue-gas components as SO_x_ and NO_x_, toxicity of amines and corrosion of the equipment [1]. A development of alternative methods for technically and economically viable CO_2_ separation is hence of considerable interest.

One of the promising approaches capable of solving this problem is CO_2_ capture via use of polymeric gas separation membranes [2]. Being a relatively new technology, membrane separation is characterized by low energy consumption, low footprint, high process stability and simplicity of equipment while high efficacy during gas separation processes [3]. However, further development of competitive techniques for application under question requires designing selective materials exhibiting both high permeability and selectivity of CO_2_ separation from light gases such as N_2_ and CH_4_ [4]. For instance, insufficient selectivity restricts practical application of membranes based on 1,2-disubstituted polyacetylenes [5] despite highest recorded permeability, thermal stability and impressive mechanical properties of these materials.

One of the known methods to improve gas transport properties of polymers is the incorporation of polar substituents as a side group of macromolecular chain. Upon such a modification, the increase in CO_2_/gas X selectivity may originate from higher affinity of polar functional groups toward CO_2_, which, in turn, leads to the increase in the CO_2_ solubility coefficient of the membrane material [6].

Among the polar substituents promising for the selective separation of CO_2_, alkylimidazolium cations are of particular interest due to high affinity of corresponding salts toward CO_2_. Indeed, there are several known studies evidencing possibility to increase CO_2_-selectivity of polymeric membranes after their modification with such substituents [7,8,9]. However, this approach has several restrictions. For instance, polymers with a lower degree of functionalization may not possess desired selectivity increase. On the other hand, excessive content of incorporated groups (depending on their chemical) may result either in loss of solubility in organic solvents suitable for membrane formation or in considerable deterioration of mechanical properties of corresponding polymer, thus making impossible to convert polymer to a practical polymeric membrane [10].

As a possible alternative route, surface modification of polymeric films is currently of particular applied and fundamental interest [11]. This approach allows one to form a layer of modified polymer of tunable thickness on the surface of the initial, mechanically durable, polymer, thus avoiding aforementioned problems. Thus, selective gas transport will only occur in thin surface layer, and ability to control the quantity of functional groups will allow yielding optimal parameters of gas separation.

In this study, a polymer belonging to the group of 1,2-disubstituted polyacetylenes, namely poly(4-methyl-2-pentyne) (PMP) was studied for surface modification. This polymer exhibits high permeability toward permanent and condensable gases as well as high selectivity in separating n-C_4_H_10_ from its mixtures with e.g., CH_4_ [12,13]. Moreover, as it comprises only C and H atoms, PMP possesses high thermal stability and low surface energy that prevents fouling and allows one to release the patterned material when used as a template [14,15,16,17,18]. Finally, due to higher stability of C-C vs. C-Si bonds [18], PMP is also more chemically stable compared to most intensively studied poly(1-trimethylsilyl-1-propyne) (PTMSP) though the latter belongs to the group of polymers with the highest known permeability coefficients of gases.

Principal possibility of butylimidazolium (BMIm) quaternization by brominated PMP (PMP-Br), i.e., replacing Br substituents of PMP with in situ formed cationic –BMIm^+^ side group (Figure 1) in the entire volume of PMP was demonstrated in [19]. However, quaternization of more than 15 mol % of initial PMP resulted in the loss of both solubility in common solvents and film-forming properties of the polymer.

Therefore, we developed a method for the incorporation of butylimidazolium bromide into the PMP macromolecules via the surface functionalization of PMP-Br^-^ films by BMIm^+^ in a medium of supercritical fluids (SCF).

Carrying out the reaction (Figure 1) in the SCFs possessing high penetrating ability into the PMP-Br can provide limited swelling and plasticization of the polymer. The latter, in turn, may result in the speedup of the quaternization process and contribute to its completeness via the intensification of mass- and heat-transfer processes. In addition, application of SCFs allows performing the process without toxic and hazardous organic solvents or with just minimal amounts necessary for membrane post-treatment. There is also an additional important advantage, namely no residual solvent needs to be removed from the polymer after the process completes [20,21].

The aforementioned advantages make SCFs widely applicable for micronization (via PGSS process, i.e., particles from Gas Saturated Solutions) [22,23], treatment of polymers [24], impregnation [25,26], etc. Besides, there are also some studies of SCFs use as a reaction media for polymerization [27] and co-polymerization [28], whereas application of SCFs for heterogeneous chemical modification of polymers is a more rare case [20].

Herein, supercritical CO_2_ and CHF_3_ which are non-toxic, non-flammable and inert toward the ozone layer have been investigated as a reaction medium for surface modification of PMP films via quaternization of BMIm by brominated polymer. As a result, we determined conditions of quaternization and content of functional groups, which are optimal for preparation of membranes suitable for CO_2_ capture from the industrial gas streams.

## 2. Materials and Methods

### 2.1. Materials

Brominated PMP was prepared according to [19]. The content of brominated moieties was determined as 26 mol %, M_w_ = 479 kg∙mol^−1^, M_w_/M_n_ = 2.

*N*-buthylimidazole (98%, Acros Organics, Nidderau, Germany) was distilled over CaH_2_ under vacuum.

CCl_4_ (chemically pure, Chimmed, Moscow, Russia) was purified by 10% aqueous solution of KOH and washed with H_2_O until the neutral reaction, then dried over anhydrous CaCl_2_ for 48 h and triple distilled over P_2_O_5_ under high-purity Ar (4.8 grade, TU 20.11.11.121–006–45905715–2017, 99.998 vol. % of Ar).

Food-grade CO_2_ (99.5%, GOST 8050-85, Linde Gas, Balashikha, Russia) and CHF_3_ (99.9%, Leona, Moscow, Russia) were used as a reaction media for quaternization of BMIm by PMP-Br.

### 2.2. Preparation of Brominated PMP Films

Polymeric membranes were prepared by pouring 1.5 mass. % solution of PMP-Br in CCl_4_ onto the leveled cellophane foil with subsequent covering by Petri dish in order to provide slow evaporation of the solvent. The films were dried at ambient conditions for 7 days followed by drying in vacuum for 48 h at 25 °C. The diameter of the obtained samples was 70 mm, and the thickness was in the 130–150 μm range.

### 2.3. Treatment of PMP-Br with BMIm in SCF Media (Quaternization of BMIm)

The experiments on quaternization of BMIm were performed on RESS/SAS equipment (Waters Corp., Plymouth Meeting, PA, USA). The scheme of the experimental setup is shown in Figure 2.

Before starting the experiment, PMP-Br films and BMIm in 1:20 molar ratio were placed into pre-heated reactor 5 (Figure 2). Then, the reactor vessel 5 (volume 380 mL) was sealed and SCF CO_2_ or CHF_3_ was injected under 350 Bar pressure provided by the pump 3. The quaternization experiment was carried out at constant temperature, which was varied from experiment to experiment in the range 40–80 °C. In all experiments, exposition time in SCF medium was 48 h. After the treatment, temperature was lowered to room temperature, and pressure was slowly relieved with valve 6. Modified polymer films were washed several times with excess of H_2_O and MeOH in order to remove residual BMIm^+^, and then dried under vacuum until constant mass.

### 2.4. Physico-Chemical Characterization

The binding of the –BMIm^+^ moieties onto the surface of the initial brominated PMP has been confirmed by attenuated total internal reflection IR spectroscopy and ATR-IR. Spectra were registered 24 times with subsequent averaging, at a resolution of 4 cm^−1^ in the 400–4000 cm^−1^ range by means of Bruker ALPHA (Bruker Optik GmbH, Ettlingen, Germany) spectrometer equipped with a diamond tool. Upon such an investigation, IR radiation of the source falls at an angle greater than the critical one on the surface of the sample tightly pressed against diamond crystal plate. This results in the reflection of the radiation from the surface of the sample, then from the inner surface of diamond while certain amount of the initial radiation penetrates the sample and is subsequently absorbed depending on the sample composition. As a result, residual reflexed radiation analyzed by the detector is also determined by the composition of the sample. Noteworthy is that the mentioned absorption of the radiation is very effective, and occurs in a very thin layer of the sample, which makes ATR-IR a perfect method to investigate surface and corresponding surface reactions, including those in polymeric films.

Mechanical properties were investigated at 20 °C at a constant stretching rate of 10 mm/min using the Instron 1122 tensile testing machine (Instron Corporation, Norwood, MA, USA). The specimen was in a form of a rectangular film with 30 mm length, 10 mm width, and a thickness ca. 140 µm depending on the sample cast from polymer solution.

The surface morphology of the polymeric films was studied by scanning electron microscopy (SEM) with Carl Zeiss NVision 40 (Carl Zeiss Microscopy GmbH, Jena, Germany) scanning electron microscope. Backscattered electron images were taken under the accelerating voltage of 1 keV and without the application of gold plating.

Thermogravimetric analysis (TGA) was performed on Mettler Toledo TGA/DSC instrument (Mettler-Toledo (Switzerland) GmbH, Greifensee, Switzerland) in the temperature range 20–1000 °C under a flow of Ar (10 mL/min). Specimen of 5–40 mg was placed into 70 μL Alundum™ crucible and then heated with a rate of 10 °C min^−1^. The accuracy of temperature measurement was ±0.3 °С, and that of mass measurements was ±0.1 μg.

Differential scanning calorimetry (DSC) measurements was performed on Mettler Toledo DSC823e (Mettler-Toledo (Switzerland) GmbH, Greifensee, Switzerland) instrument in the temperature range 20–350 °C under a flow of Ar (70 mL/min). Five to twenty milligrams of the sample was placed into 40 μL Alundum™ crucible, which was then closed with a perforated lid thus allowing carrying out measurement at constant (atmospheric) pressure. The heating rate was 20 °C/min. The experimental data were processed with STARe software supplied with the instrument. The temperature determination was ±0.2 °С.

In order to evaluate the solubility of samples in solvents, ≈100 mg of each sample was placed into the glass vial with 25 mL of corresponding solvent and left at 25 °C for 24 h, then heated to 60 °C for 6 h. After that, the solution was filtered and the filtrate was precipitated into methanol. The solubility of the polymer was determined by the difference in weights of initial and precipitated samples.

The permeability of the films toward particular gases was studied at 30 °C on a laboratory setup designed to measure the parameters of gas permeability of flat films working according to the “constant volume/variable pressure” principle [29]. Permeability coefficients (P) for membranes of known thickness were calculated as: (1)$$P=D\times S=\frac{V_{p}l}{ART∆t}ln\frac{p_{f}-p_{p1}}{p_{f-p_{p2}}}$$
where *V_p_* is the permeate volume, *l* is the film thickness (130–150 μm), *A* is the surface area of the membrane, *R* is the universal gas constant, $p_{f}$ is the feed pressure of gas applied to the membrane (1 bar for all the gases in the Δt time interval), $p_{p1}$ and $p_{p2}$ are permeate pressure at times 1 and 2, Δt is the difference in time between points 1 and 2. The error of permeability coefficient value determination did not exceed 5%.

The ideal selectivity was calculated as a ratio of permeability coefficients of individual gases A and B:(2)αAB=PAPB
where αAB is ideal selectivity, $P_{A}$ and $P_{B}$ are permeability coefficients for gases А and B.

## 3. Results

For the preparation of butylimidazole-containing derivatives of PMP, PMP-Br films were treated by BMIm in SCF CO_2_ or CHF_3_ medium at various temperatures. The summary of experimental conditions is presented in Table 1.

Figure 3 represents visual changes of the films upon the quaternization. Gradual change of color in the series of samples 1–3 and 4–6 indicates an increasing degree of modification with an increase of temperature of treatment. Obviously, sample 6 treated in the CHF_3_ has more intensive color in comparison to sample 3 treated in the CO_2_, thus clearly indicating SCF preferable for PMP-Br quaternization.

It is worth noting that polymer comprising 26 mol % of brominated moieties was found to be the most optimal for studied modification since polymers with a higher Br content are not soluble in solvents commonly applied for the preparation of films.

The obtained results evidence the occurrence of quaternization, i.e., substitution of Br atoms by –BMIm^+^ groups in PMP-Br films (Figure 1). Both the process temperature and nature of the supercritical fluid affected the degree of quaternization.

Indeed, the IR spectra of samples with assumed Br to BMim^+^ substitution (Figure 4 and Figure 5) demonstrate considerable changes in the intensive complex band near 618 cm^−1^. Namely, a decrease in the intensity of a shoulder at 630 cm^−1^ in IR spectra of samples 1–3 makes the profile of this band more symmetric compared to that of the PMP-Br. Since valence vibrations of C-Br bonds, ν_C-Br_, are known to be found in this range [30], the described changes do originate from the decrease in the amount of Br groups during the quaternization process.

Besides, higher quaternization degree of samples 2 and 3 compared to that of sample 1 can be assumed since IR spectra of samples 2 and 3 do not comprise distinct bands at ≈840 cm^−1^ (Figure 4b) and ≈1240 cm^−1^ (Figure 4c), respectively, corresponding to various bending C–H vibrations and δ_C–H_ [30]. The disappearance of these bands is apparently due to their considerable shift and merging with other, more intense bands. The latter, in turn, originates from considerable change in nature of peripheral substituents, i.e., from Br to –BMIm^+^.

In addition to changes common for IR spectra of all the quaternized samples, one can conclude on higher efficiency of SCF CHF3 comparing to SCF CO_2_ toward the quaternization process. Indeed, in IR spectra of corresponding samples 4–6, only one distinct band at 620 cm^−1^ without any νC-Br shoulders at 630 cm^−1^ is discernible (Figure 5a), and δ_C–H_ bands at 840–850 and 1240 cm^−1^ are shifted in all the spectra (Figure 5b,c). Among 4–6, in turn, the highest quaternization degree can be assumed for sample 6 due to the absence of sufficiently intensive distinct or even shoulder-like δ_C–H_ band of the initial PMP-Br at 1240 cm^−1^ (Figure 5c).

Since IR examination have been performed similarly in all the cases, authors believe that changes in intensities as well as broadening and splitting of particular bands in the IR spectrum of the sample 6 originates from comparatively higher content of H_2_O involved in interactions with polar BMIm substituents and, possibly, Br-. A higher H_2_O content in sample 6 is clearly evidenced as a very broad band at 3300 cm^−1^ (see Appendix A) not registered in IR spectra of other samples.

SEM investigation of the film surface before and after quaternization (Figure 6) showed a change in the film surface morphology, which can be attributed to a limited swelling of the PMP-Br in SCF CO_2_ leading to leveling out of artefacts of film preparation. Unfortunately, due to limitations to experimental work caused by the COVID-19, it was not possible to investigate cross-sectional morphology of quaternized films and the depth of quaternization. One can only speculate that, in case both SCFs are swelling agents for the PMP-Br, during the 48 h long exposure of films to SCF, BMIm can diffuse into the depth of the PMP-Br film through more acessible free volume voids and cause formation of quaternized material through the whole depth of the film at least in the vicinity of aforementioned free volume voids.

Film-forming, strength and relaxation properties of the quaternized PMP films were investigated. All samples have kept flexibility (Figure 7) after processing under any of the tested conditions. Besides, strength characteristics of quaternized polymeric films are similar to those of the initial polymer (Young’s modulus of 1000 MPa, tensile strength at break of 45 MPa, strain at break of 50%). These results evidence that polymer modification does not lead to the deterioration of strength characteristics of polymer. The latter, in turn, determines the principal possibility of applying a modified polymer as a membrane material.

It is also important to mention that modified polymeric films retain high thermal stability typical for disubstituted polyacetylenes comprising bulky substituents. Indeed, as evidenced in Figure 8, initial PMP and quaternized sample **6** start to decompose (5% of mass loss) at mostly the same temperature: 230 and 210 °C, respectively. The results of TGA investigation of all 6 quaternized samples are presented in Appendix A.

The results of DSC study evidence the absence of thermal effects corresponding to glass transition and fluidity up to the temperature of polymer decomposition for all studied samples. This fact allows one to conclude that, similar to non-modified PMP homopolymers, temperatures of these transitions lie above that of the onset of chemical decomposition. Examples of DSC measurement results are presented in the Appendix A.

Besides the thermal stability, resistance against various organic compounds is also essential for the possible application as membrane materials for CO_2_ separation. All the prepared quaternized PMP films retain stability against aliphatic and aromatic hydrocarbons, a property, which is characteristic for both PMP and PMP-Br polymers.

As it was mentioned in the experimental part, water and methanol were used for washing the modified films after exposure to SCF. Both these liquids are non-solvents for all studied polymers and the quaternization reaction was possible neither in these liquids nor in other organic solvents, e.g., dimethylsulfoxide (DMSO) able to dissolve BMIm but not dissolve the PMP-Br.

The study of the gas transport properties of the obtained polymeric films revealed considerable superiority of modified polymers over the initial PMP in CO_2_/(gas X) selectivity, namely by 2.5–3 and 2–4 times for the CO_2_/N_2_ and CO_2_/CH_4_ gas pairs, respectively (Table 2). Let us also emphasize on retaining high values of the permeability coefficients at the same time. Such an increase in CO_2_-selectivity is apparently due to stronger interactions of ionic –BMIm^+^ and Br^-^ moieties with CO_2_ molecules compared with similar interactions involving non-polar CH_4_ and N_2_. Intoduction of the bulky BMim^+^ group into the structure of the polymer being already in solid but possibly swollen state should result in an alteration of the free volume voids arrangement, presumably decreasing the free volume void size, which should result in an increase of at least diffusivity selectivity. Unfortunately, it is not possible at this stage of research to investigate the free volume of obtained polymers by any physical method.

The chosen approach of polymer modification has one significant advantage, namely chemical modification of the polymer only in zones, where, presumably, the free volume elements responsible for small molecule transport are accessible for the modification agent. The rest of the polymer remains intact keeping properties of the polymer not related to transport at the level of the initial material. Since SCF causes swelling of the polymer, relatively large BMIm molecules can penetrate into the depth of the polymer matrix along the gas diffusion path.

Analysis of the permeability coefficients’ dependence on quaternization reaction temperature (Table 1 and Table 2) shows that the saturation of the PMP-Br with quaternized BMIm was not finished even after exposure to 80 °C for 48 h (Figure 9). This fact is of particular interest because diffusion coefficients known for disubstituted polyacetylenes and bulky penetrant, e.g., n-C_4_H_10_, are in the range of 1 × 10^−7^ cm^2^ s^−1^ at 30 °C and increase with temperature rise. This is because the sample with a 150-µm-thickness exposure for 48 h to BMIm should be more than sufficient for equilibration of BMIm concentration within the polymer film. Since no stabilization of gas transport properties was observed, one can speculate that consumption of BMIm for quaternization reaction results in the free volume alteration and thus restricts transport of the BMIm into the depth of the polymer. Further experiments with various exposure times and temperatures are necessary.

The aforementioned alteration of the free volume can be the reason for very small aging observed for the quaternized sample 6. Samples of PMP-Br and sample 6 were tested for gas transport properties changes caused by either treatment in SCF or aging. Data presented in Table 2 clearly shows that PMP-Br in 48 h after exposure to the supercritical CO_2_ demonstrates permeability coefficients higher than that of non-treated PMP-Br film, and only after 4 month aging time degrades in gas transport properties to the level of the initial PMP-Br both in terms of permeability and selectivity. Sample 6 was investigated for the gas transport properties according to the same procedure and after 4 month demonstrated stability of CO_2_ permeability coefficient and some increase in CO_2_/N_2_ and CO_2_/CH_4_ selectivity. One can attribute this very minor change of properties to the restriction of relaxation process in the polymer into which bulky ionic functional groups were introduced while polymer was already in the solid state.

A conventional method for assessing the quality of the gas separation properties of a particular membrane material is its location relative to the “Upper bound” of the Robeson diagram representing the trade-off between the permeability coefficients and ideal selectivity for various polymers [31].

Namely, the closer the polymer is to the “Upper bound”, the higher result in the separation process can be achieved in both investment (permeability coefficient converts into membrane permeance and finally into the required membrane area) and operation costs (selectivity of the membrane will be related to the energy necessary for compression and recirculation of gas streams).

Figure 10 and Figure 11 demonstrate trade-off between CO_2_/N_2_ and СO_2_/CH_4_ ideal selectivities and CO_2_ permeability coefficients for a variety of polymers [31,32]. As one can see, PMP comprising maximal amount of –BMIm^+^ moieties (sample 6) is located near the upper border of the diagram in both cases, and exhibits more attractive gas separation properties compared to the initial PMP. These results allows one to consider sample 6 as a promising material for the separation of CO_2_-containing gas mixtures.

Modeling of the membrane gas separation process of CO_2_ removal from flue gas of hard coal power plant [33,34] shows that in case thin film composite (TFC) membrane will have a selective layer of sample 6 with 100 nm effective thickness, which is a common value for PolyActive™ TFC membrane, the CO_2_ concentration in the permeate for the sample 6 will exceed 50 mol % compared to 64 mol % for PolyActive™ but necessary membrane area will be 10 times smaller. The CO_2_ content above 50 mol % is sufficient for the second membrane separation stage where less permeable but more selective membrane will finally concentrate CO_2_ to >95 mol % required for liquefaction. These facts as well as chemical stability of the quaternized PMP-BMIm make the developed materials interesting for further investigation in real gas separation environment.

## 4. Conclusions

To sum up, we demonstrated the principal possibility and efficiency of applying supercritical CO_2_ and CHF_3_ as a reaction medium for surface modification of PMP films for quaternization of BMIm by brominated polymer. While different treatment temperatures were examined, the results of comparative IR study evidences that maximal quaternization degree is achieved during polymer processing in CHF_3_ at 80 °C, the highest studied temperature. The resulting quaternized PMP films retain good mechanical strength as well as high thermal and chemical stability of the initial polymer combined with increased CO_2_-selectivity while maintaining the permeability coefficients at a high level. Such a set of properties makes these modified polymers promising membrane materials for CO_2_ removal from the industrial gas mixtures containing hydrocarbons or from the flue gas streams.

Besides, our results evidence the prospects for further development of surface modification of PMP via treatment in SCF medium for preparation of membrane materials with increased selectivity of CO_2_ separation.

## Figures and Tables

**Figure 1 polymers-12-02468-f001:**
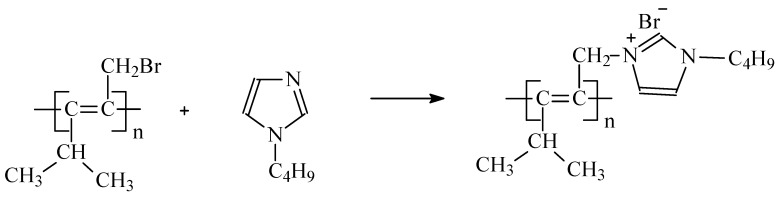
Quaternization of butylimidazolium (BMIm) by brominiated poly(4-methyl-2-pentyne) (PMP-Br).

**Figure 2 polymers-12-02468-f002:**
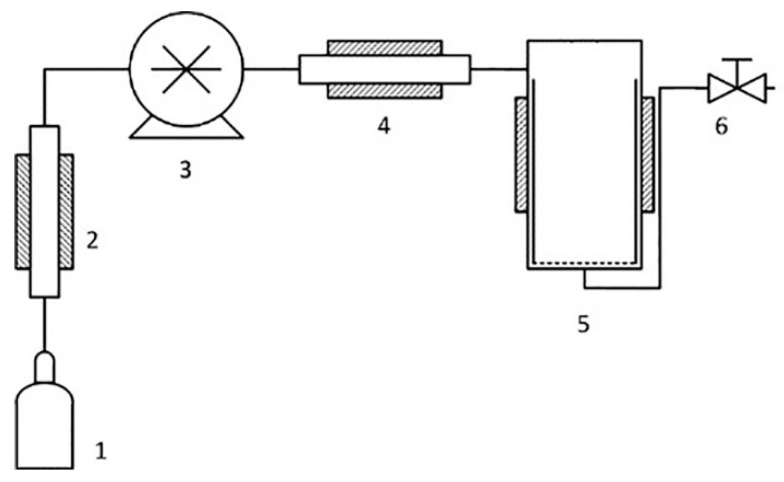
Equipment for the BMIm quaternization by PMP-Br in supercritical fluids (SCF) (1, source of CO_2_ or CHF_3_; 2, cooling heat exchanger; 3, high-pressure pump; 4, heater; 5, reactor; 6, pressure relief valve).

**Figure 3 polymers-12-02468-f003:**
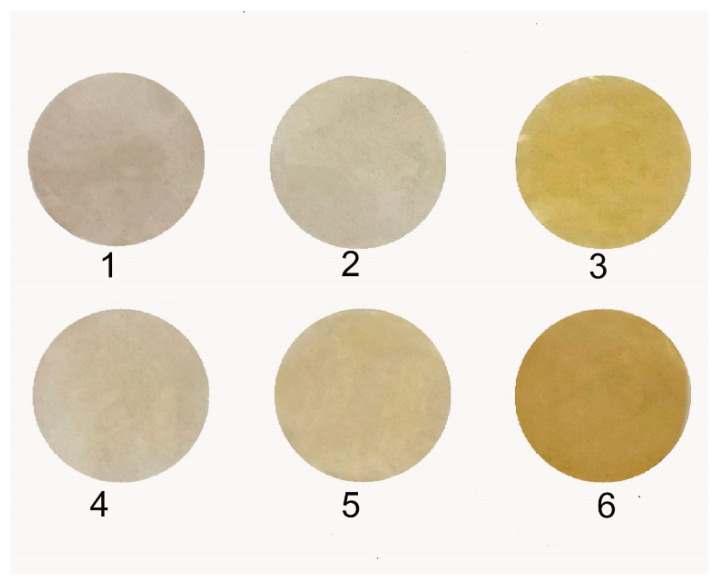
Changes in visual appearance of quaternized samples 1–6.

**Figure 4 polymers-12-02468-f004:**
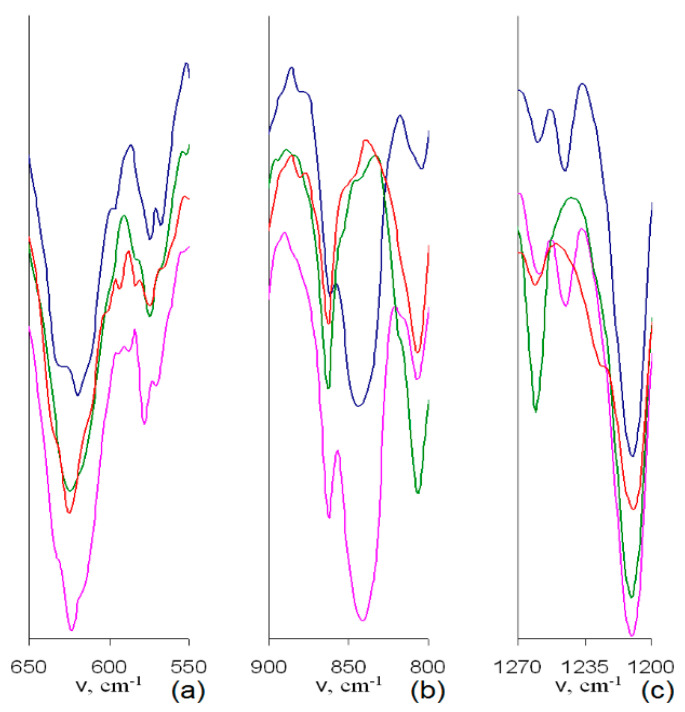
Normalized IR spectra of PMP-Br (blue line), samples 1 (pink line), 2 (green line) and 3 (red line) in the range of 650–550 cm^−1^ (**a**), 900–800 cm^−1^ (**b**) and 1270–1200 cm^−1^ (**c**).

**Figure 5 polymers-12-02468-f005:**
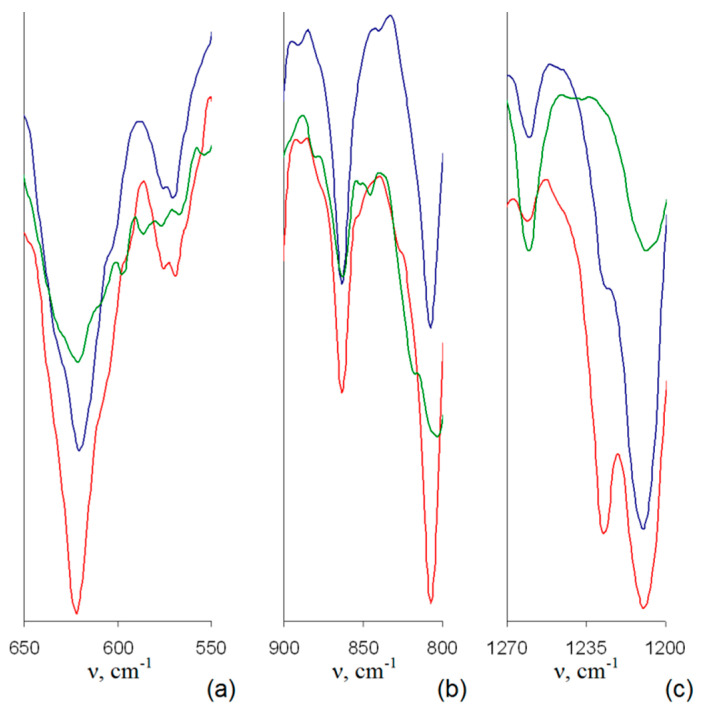
Normalized IR spectra of the samples: 4 (blue line), 5 (red line), and 6 (green line) in the range of 650–550 cm^−1^ (**a**), 900–800 cm^−1^ (**b**) and 1270–1200 cm^−1^ (**c**).

**Figure 6 polymers-12-02468-f006:**
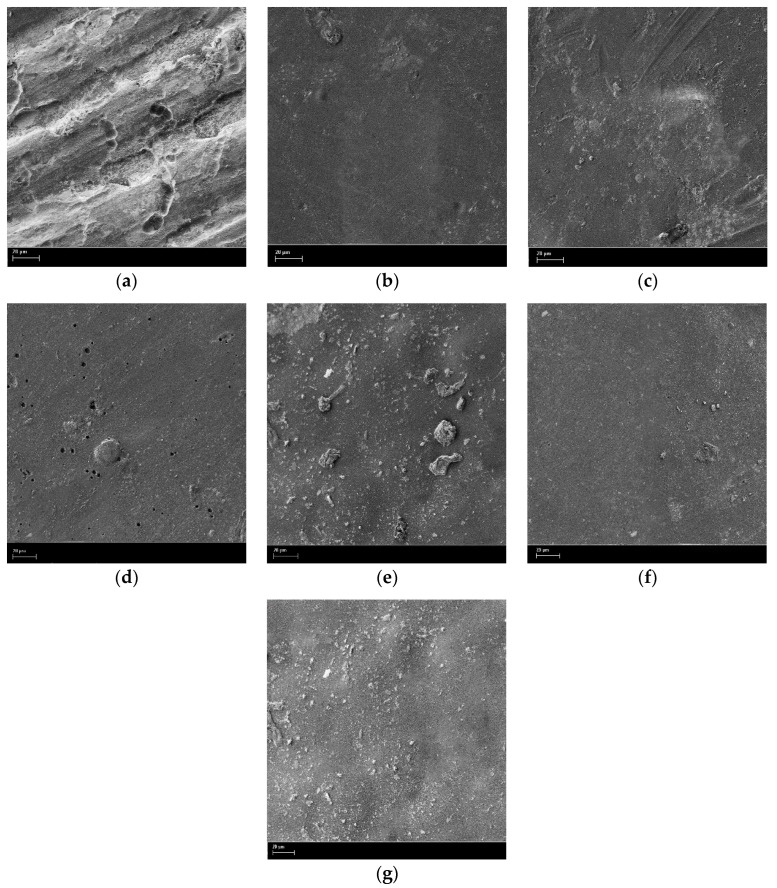
SEM images of the film surface of PMP-Br (**a**) and quaternized samples: 1 (**b**), 2 (**c**), 3 (**d**), 4 (**e**), 5 (**f**), and 6 (**g**).

**Figure 7 polymers-12-02468-f007:**
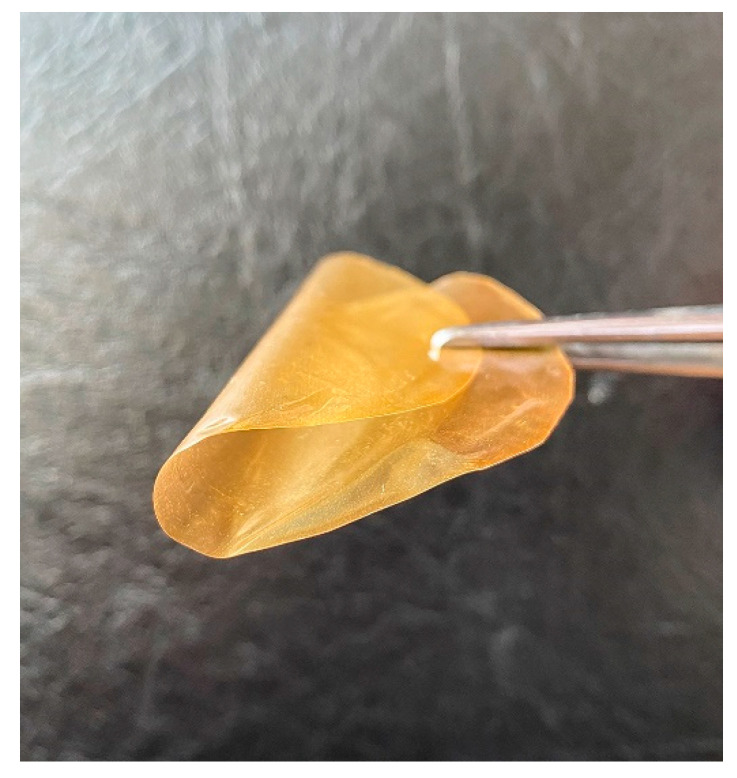
Flexibility of the quaternized PMP (sample 6).

**Figure 8 polymers-12-02468-f008:**
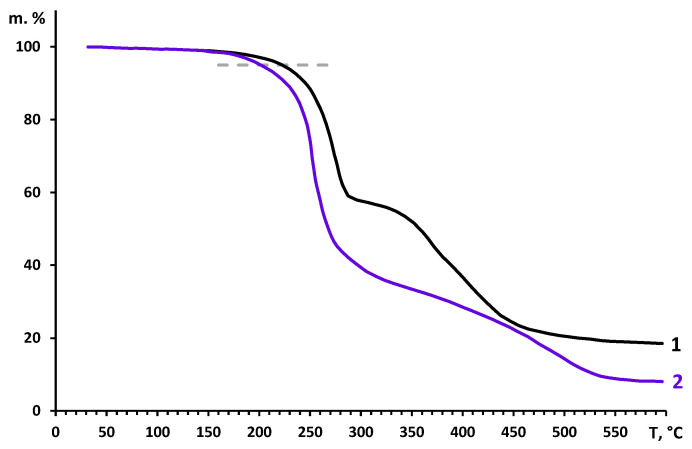
TGA curves for films of initial PMP (1) and quaternized sample **6** (2).

**Figure 9 polymers-12-02468-f009:**
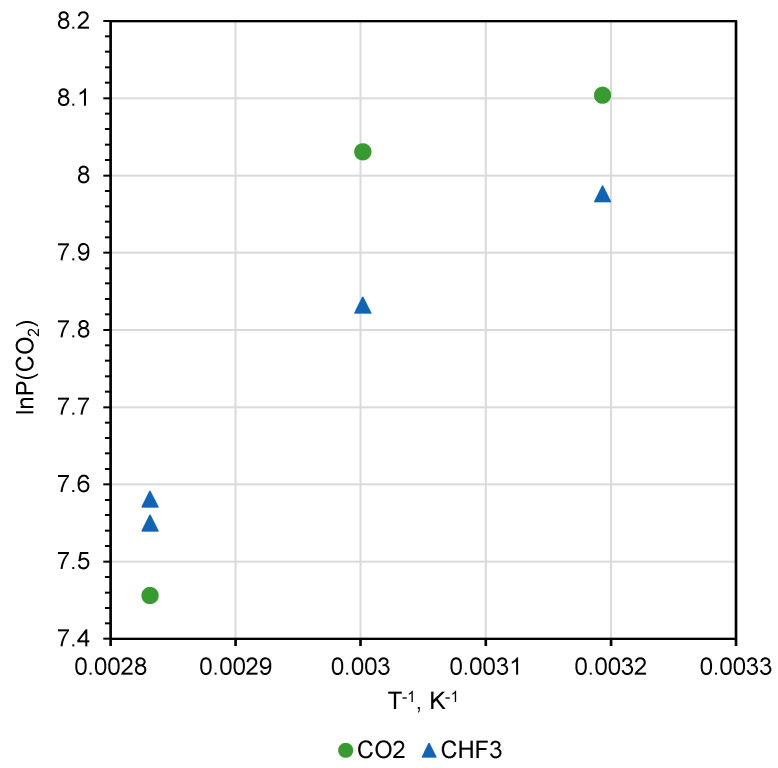
Arrhenius type representation of CO_2_ permeability coefficients in dependence on temperature of quaternization reaction in two SCFs.

**Figure 10 polymers-12-02468-f010:**
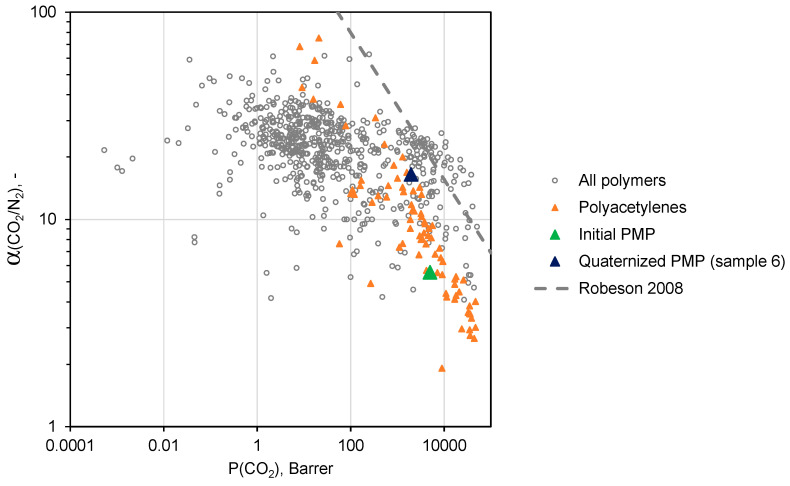
Robeson diagram for CO_2_/N_2_ mixtures.

**Figure 11 polymers-12-02468-f011:**
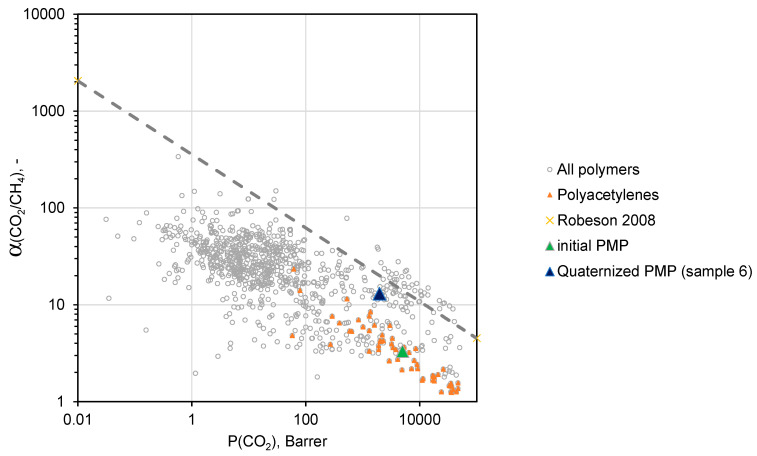
Robeson diagram for CO_2_/СH_4_ mixtures.

**Table 1 polymers-12-02468-t001:** Conditions for the quaternization of BMIm by PMP-Br films in the medium of CO_2_ and CHF_3_ SCFs.

Sample	Pressure, Bar	BMIm Volume, mL	Time, h	SCF	T, °С
1	350	15	48	CO_2_	40
2	”*	”	”	”	60
3	”	”	”	”	80
4	350	15	48	CHF_3_	40
5	”	”	”	”	60
6	”	”	”	”	80

* The mark (”) indicates “same as above”.

**Table 2 polymers-12-02468-t002:** Permeability coefficients and ideal selectivities of the initial and modified PMP films toward individual gases.

Sample	Permeability (P), Barrer ^1^			Selectivity (α)	
	CO_2_	N_2_	CH_4_	P(CO_2_)/P(N_2_)	P(CO_2_)/P(CH_4_)
PMP	5000	900	1500	5.6	3.3
PMP-Br	4800	650	1000	7.4	4.8
PMP-Br ^2^	5200	700	1200	7.4	4.3
PMP-Br ^3^	4700	630	950	7.5	4.9
1	3307	271	541	12.2	6.1
2	3073	217	430	14.2	7.1
3	1730	111	171	15.6	10.1
4	2910	230	438	12.7	6.7
5	2520	165	328	15.3	7.7
6 ^4^	1960	119	159	16.5	12.3
6 ^5^	1900	112	150	17.0	12.7

^1^ 1 Barrer = 10^–10^ cm^3^(STP) × cm × cm^–2^ × s^–1^ × cmHg^–1^. ^2^ 48 h after SCF CO_2_ treatment. ^3^ 4 month after SCF CO_2_ treatment. ^4^ 48 h after SCF CHF_3_ treatment. ^5^ 4 month after SCF CHF_3_ treatment.

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
