# Peer review of "Modification of Poly(4-methyl-2-pentyne) in the Supercritical Fluid Medium for Selective Membrane Separation of CO2 from Various Gas Mixtures"

_polymers, 2020, doi:10.3390/polym12112468_

Round 1

Reviewer 1 Report

The authors shows the brominated poly(4-methyl-2-penthine) to separate CO2/N2. The novel membrane has the separation ability up to 17 (CO2/N2) and 12.6 (CO2/CH3). Although the detailed mechanism for chemical substitute is not efficient, the approach is quite valuable for bromination. Therefore, I expect if the authors will try to design the membrane structure more detailed, more enhanced performances would be shown.

Author Response

Reviewer 1

Comments and Suggestions for Authors

The authors shows the brominated poly(4-methyl-2-penthine) to separate CO2/N2. The novel membrane has the separation ability up to 17 (CO2/N2) and 12.6 (CO2/CH3). Although the detailed mechanism for chemical substitute is not efficient, the approach is quite valuable for bromination. Therefore, I expect if the authors will try to design the membrane structure more detailed, more enhanced performances would be shown.

Answer: The detailed description of processes occurring during the brominated PMP modification in the medium of supercritical fluid are to be investigated by molecular modelling and observation of interaction of penetrants of various nature with quaternized polymer. So far, the hypothesis is that limited swelling caused by SCF allows BMIm to penetrate into the free volume voids (FVV) of PMP-Br where “surface” reaction is happening. By the “surface” authors understand parts of PMP-Br surrounding the FVV, the depth of BMIm penetration into the FVV wall is not clear and authors are looking for a physical method able to resolve chemical composition of synthesized compound with resolution in the range <1 nm. High-resolution transmission electron microscopy can be of help, Time-of-flight secondary ion mass spectrometry having depth resolution of 1nm but, unfortunately, lateral resolution in the range 400-2000 nm is of interest as well.

Reviewer 2 Report

The manuscript "Modification of poly(4-methyl-2-pentine) in the supercritical fluid medium for selective membrane separation of CO2 from various gas mixtures" by V. Polevaya et al. presents the synthesis of a new membrane designed for the CO2 separation from gases mixtures. In particular the authors reports the modification of highly permeable films of brominated 1,2-disubstituted polyacetylene, poly(4-methyl-2-penthine) by means of quaternization of butylimidazolium applying supercritical CO2 and CHF3 as a reaction medium for surface modification.

The idea is interesting from both a fundamental and an applicative point of view, and the research is well planned, since several experimental measurements are used to characterize the samples obtained by tuning dfferent parameters of the modification procedure. However in my opinion the experimental results are not clearly described and discussed, therefore several major revisions are necessary in order to publish the paper. I will detail my observation n the following:

  • Table 1 reporting the details of sample preparation is confusing and should be changed.
  • IR experiments: at lines 315 and 316 the authors affirm that the sample obtained at higher temperature in supercritical CHF3 (sample 6) has the maximum amount of imidazolium units, I think this conclusion is suggested by the IR results but it is not properly discussed at the end of IR data presentation (lines 221-225). Moreover the authors should comment why sample 6 displays the lowest intensity spectrum (or are the data plotted in arbitrary units? This point should be clarified)
  • SEM pictures of only one sample is not enough to draw any conclusion. At least the surface images of all the samples could be added.
  • Mechanical properties (lines 238-244): how the reported vales have been obtained? I could not find experimental details neither graphs of the performed measurements, except for a picture.
  • DSC measurements are not displayed
  • TGA measurements: I suggest to show TGA data of all the synthesized samples to compare the effects of the different preparation procedures. This is true for all the experimental techniques used to characterize the obtained samples.
  • Results in table 2 should be better discussed, in particular the permeability values of the different samples should be clearly compared.

Author Response

Reviewer 2

Comments and Suggestions for Authors

The manuscript "Modification of poly(4-methyl-2-pentine) in the supercritical fluid medium for selective membrane separation of CO2 from various gas mixtures" by V. Polevaya et al. presents the synthesis of a new membrane designed for the CO2 separation from gases mixtures. In particular the authors reports the modification of highly permeable films of brominated 1,2-disubstituted polyacetylene, poly(4-methyl-2-penthine) by means of quaternization of butylimidazolium applying supercritical CO2 and CHF3 as a reaction medium for surface modification.

The idea is interesting from both a fundamental and an applicative point of view, and the research is well planned, since several experimental measurements are used to characterize the samples obtained by tuning dfferent parameters of the modification procedure. However in my opinion the experimental results are not clearly described and discussed, therefore several major revisions are necessary in order to publish the paper. I will detail my observation n the following:

  • Table 1 reporting the details of sample preparation is confusing and should be changed.

Answer: The Table 1 is changed, hopefully it is more understandable in the new form.

  • IR experiments: at lines 315 and 316 the authors affirm that the sample obtained at higher temperature in supercritical CHF3 (sample 6) has the maximum amount of imidazolium units, I think this conclusion is suggested by the IR results but it is not properly discussed at the end of IR data presentation (lines 221-225). Moreover the authors should comment why sample 6 displays the lowest intensity spectrum (or are the data plotted in arbitrary units? This point should be clarified)

Answer: We can assume the highest degree of Br to BMIm+ substitution in the case of sample 6 since IR spectrum of this sample does not comprise neither distinct nor even shoulder-like δC-H band of sufficiently intensity at 1240 cm-1 corresponding to the initial PMP-Br. Corresponding alterations are made in the main text.

We apologize for poor explanation in the initial Manuscript, this remark concerned only the areas of the IR spectra being discussed and plotted in arbitary units. In general, IR spectrum of Sample 6 exhibits uneven changes in intensities as well as broadening and splitting of certain bands. We believe these changes originate from comparatively higher content of H2O involved in interactions with polar BMIm+ substituents and, possibly, Br-. In order to more adequately compare IR data, normalization of spectra was performed, and Figures S1 and S2 of the Supporting Information showing normalized spectra in the entire region have been added and breefly discussed in the Footnote 1.

  • SEM pictures of only one sample is not enough to draw any conclusion. At least the surface images of all the samples could be added.

Answer: SEM images of the film surface for all samples are added to the Figure 6. As one can observe the visual roughness of samples exposed to SCF is significantly reduced compared to as cast film of PMP-Br.

  • Mechanical properties (lines 238-244): how the reported vales have been obtained? I could not find experimental details neither graphs of the performed measurements, except for a picture.

Answer: Description of mechanical testing is included into the manuscript.

  • DSC measurements are not displayed

Answer: Examples of 3 DSC measurements are included into the Supporting information. No glass transition can be observed in the studied temperature range.

  • TGA measurements: I suggest to show TGA data of all the synthesized samples to compare the effects of the different preparation procedures. This is true for all the experimental techniques used to characterize the obtained samples.

Answer: TGA measurements showed no significant difference in the temperature of polymer degradation, all quaternized samples have start of decomposition within 10°C range. Authors would like to insist on the Figure 8 in the current form since it clearly shows difference in degradation curves between PMP-Br and one of quaternized samples, namely sample 6 which was studied in detail.

  • Results in table 2 should be better discussed, in particular the permeability values of the different samples should be clearly compared.

Answer: The discussion on Table 2 is improved and included into the text. Special emphasis is made on aging of PMP-Br and Sample 6. PMP-Br shows very limited aging during 4 month after sample exposure to supercritical fluid, and sample 6 has even better aging characteristics which authors relate to restrictions to relaxation in the vicinity of free volume voids caused by appearance of bulky ionic moieties in the structure of the polymer on “walls” of free volume elements.

Round 2

Reviewer 2 Report

I appreciated all the changes introduced by the authors to address my comments. I still believe that the TGA figure should be commented more, at least describing the modification of the lines and commenting in the text the decomposition temperatures of other samples. The TGA data of all the synthesized samples could be at least added to the SI.

Author Response

Authors are thankful to the reviewer for valuable comments, which allowed to improve the manuscript!

The Figure S4 was added to the supporting information file. It contains TGA curves for all 6 quaternized samples.

The Figure 8 of the main text was adjusted since in the previous version of the manuscript, due to some error, the legend of the Y-Axis was somewhat shifted. Authors apologize for this mistake in representation of graphical information!